# Mechanical stiffness of reconstituted actin patches correlates tightly with endocytosis efficiency

Jessica Planade[1]☯, Reda Belbahri[1,2]☯, Micaela Boiero Sanders[2], Audrey Guillotin[2], Olivia du Roure[1]*, Alphée Michelot[2]*, Julien Heuvingh[1]*

**1** Physique et Mécanique des Milieux Hétérogènes (PMMH), ESPCI Paris, PSL University, CNRS UMR 7636, Université Paris Diderot, Sorbonne Université, Paris, France, **2** Aix Marseille Université, CNRS, IBDM, Turing Centre for Living Systems, Marseille, France

☯ These authors contributed equally to this work.
* olivia.duroure@espci.fr (OdR); alphee.michelot@univ-amu.fr (AM); julien.heuvingh@espci.fr (JH)

**Data Availability Statement:** All relevant data are within the paper and its Supporting Information files.

## Abstract

Clathrin-mediated endocytosis involves the sequential assembly of more than 60 proteins at the plasma membrane. An important fraction of these proteins regulates the assembly of an actin-related protein 2/3 (Arp2/3)-branched actin network, which is essential to generate the force during membrane invagination. We performed, on wild-type (WT) yeast and mutant strains lacking putative actin crosslinkers, a side-by-side comparison of in vivo endocytic phenotypes and in vitro rigidity measurements of reconstituted actin patches. We found a clear correlation between softer actin networks and a decreased efficiency of endocytosis. Our observations support a chain-of-consequences model in which loss of actin crosslinking softens Arp2/3-branched actin networks, directly limiting the transmission of the force. Additionally, the lifetime of failed endocytic patches increases, leading to a larger number of patches and a reduced pool of polymerizable actin, which slows down actin assembly and further impairs endocytosis.

## Introduction

Endocytosis is a key process that regulates the internalization of extracellular material and the homeostasis of the plasma membrane. Clathrin-mediated endocytosis, which is the main pathway to endocytosis in yeast, has been characterized as a multistage process implicating more than 60 different proteins [1–3]. The process begins with the assembly at the membrane of an inner layer coat of adaptor proteins, among which clathrins form a scaffold for the forming vesicle. Nucleation promoting factors of actin assembly are later recruited at endocytic sites before the assembly of a network of actin filaments branched by the actin-related protein 2/3 (Arp2/3) complex [1,4,5]. These actin networks, which surround the forming vesicle, have a diameter of about 200 nm and therefore appear as small patches at the resolution of a fluorescence microscope [6]. Actin polymerization at the plasma membrane is essential in yeast to counteract the high turgor pressure present in these cells and to provide the force necessary for the deformation of the membrane and its internalization [7,8]. Any perturbation brought

**Funding:** This project has received funding from the European Research Council (ERC) under the European Union's Horizon 2020 research and innovation program (grant agreement n˚ 638376/ Segregactin), from the Labex INFORM (ANR-11-LABX-0054, funded by the 'Investissements d'Avenir French Government program') and from the French Agence Nationale de la recherche (ANR), under grant ANR ANR-15-CE13-0004 (MuScActin). We acknowledge the France-BioImaging infrastructure, which is also supported by the ANR (ANR-10-INSB-04-01). The funders had no role in study design, data collection and analysis, decision to publish, or preparation of the manuscript.

**Competing interests:** The authors have declared that no competing interests exist.

**Abbreviations:** Abp140, actin-binding protein 140; Arp2/3, actin-related protein 2/3; CH, calponin homology; CLR, calponin-like repeat; DoG, difference of Gaussian; GFP, green fluorescent protein; Las17, yeast homolog of WASp; NPF, nuclear promoting factor; PRD, proline-rich domain; Sac6, yeast homolog of fimbrin; Scp1, yeast homolog of calponin; WASp, Wiskott-Aldrich syndrome protein; WT, wild type.

to actin assembly, for example, by the addition of latrunculin A, has dramatic effects on the rate and on the efficiency of endocytosis [9].

Actin patch assembly is correlated with the recruitment of numerous accessory proteins which decorate actin filaments [2]. These proteins affect actin filament organization and dynamics and are crucial in providing actin networks with optimized properties for efficient endocytosis. Among these accessory proteins, actin crosslinkers have been identified as a potential mechanical linkage between actin filaments in addition to the Arp2/3 complex branches. In budding yeast, three putative actin crosslinkers have been identified and localized to actin patches [2]: 1) the yeast homolog of fimbrin (Sac6), which is composed of four tandem calponin homology (CH) domains organized in two distinct actin-binding regions; 2) the yeast homolog of calponin (Scp1), which is composed of an N-terminal CH domain, a proline-rich domain (PRD), and a C-terminal calponin-like repeat (CLR). Calponin actin bundling occurs through two separate actin-binding regions in the PRD and CLR regions; 3) actin-binding protein 140 (Abp140), which is a much less studied yeast-specific actin-bundling protein. Sac6 is reported to be an important protein for endocytosis in yeast, particularly for the initiation of membrane bending and for reaching scission stage [10]. On the contrary, absence of Scp1 or Abp140 causes no obvious change in cells. However, a strong genetic interaction between Sac6 and Scp1 highlights a partially redundant function of these two proteins [11,12].

The impact of crosslinkers on the mechanics of entangled actin filaments has been extensively studied by rheometry in the past decades [13–15]. Creating permanent bonds between entangled filaments drastically increases the elastic moduli from a value on the order of 1 Pa to a value in the 100 Pa vicinity. More recently, several teams have been able to measure the mechanics of reconstituted branched actin networks that are polymerized from a surface and crosslinked by the Arp2/3 complex [16–19]. These networks, which are closer to the ones present in yeast endocytosis, differ from entangled filaments in their much higher density because of the growth process that occurs from a surface. They also already possess mechanical bonds between their filaments (the Arp2/3 complex), although each of these bonds is connected to three strands of filament instead of four in the case of crosslinkers. In the case of endocytic actin patches, it is not clear where crosslinking of filaments occurs within a highly branched network of actin filaments, in which the average branch-to-branch distance is 50 nm (which corresponds to only 20 actin subunits) [20]. Bieling and colleagues showed a moderate effect of crosslinkers on branched networks grown from a mix of purified proteins [17]. The strongest effect was obtained on the linear elasticity in presence of alpha-actinin or filamin (3.1 kPa for both to be compared to 1.6 kPa without crosslinkers). To our knowledge, no measurement has been conducted before this study on the effect of crosslinkers on networks reconstituted from cell extracts.

The link between actin networks' elasticity and their ability to invaginate the membrane is still unclear. The general idea is that a soft actin network will inefficiently transmit the polymerization force necessary to bend the membrane and surpass turgor pressure [8]. While many theoretical models have been proposed to describe force production in endocytosis [8,21], very few take explicitly into account the rigidity of the actin meshwork. The exception is the work of Tweten and colleagues [22], which uses a continuous mechanics approach to model actin growth leading to endocytosis. In this model, the actin network cannot efficiently produce endocytosis if the elastic modulus is below 80 kPa. Beyond the rigidification that allows for the transmission of force, crosslinkers could possibly store in their deformation elastic energy that could be released to aid endocytosis. Computational modeling demonstrates that because of the twisting of actin filaments, fimbrin crosslinkers could store as much as one-sixth of the energy needed for endocytosis [23]. Additional elastic energy could be also stored in the bending of filaments to be released upon endocytosis through crosslinkers unbinding.

While actin crosslinkers rigidify actin networks in vitro, recent genetics and cell biological studies suggest additional effects of these proteins in cells. Binding of fimbrin to actin filaments is competitive with other actin-binding proteins such as tropomyosin [24,25]. Consequently, loss of function of fimbrin in fission yeast correlates with a mislocalization of tropomyosin to actin patches, therefore complicating the interpretation of endocytic phenotypes [24]. Indeed, multiple direct and indirect effects of actin crosslinkers make their precise contribution to clathrin-mediated endocytosis difficult to isolate. Our goal here is to achieve a better understanding of endocytosis through the combination of phenotypic observations in cells and direct measurements of the mechanical properties of actin patches in the presence or in the absence of these crosslinkers. In this study, we took advantage of the possibility to reconstitute endocytic actin patches from yeast protein extracts [26]. We combined this approach with a high-throughput mechanical measurement technique using chains of magnetic microbeads [27]. We used a top-down approach to compare the mechanical properties of actin patches assembled from various mutant yeast strains with the corresponding endocytic defects in cells.

## Results

### Three putative actin crosslinkers impact differently membrane invagination during clathrin-mediated endocytosis

Previous studies demonstrated that a careful study of actin patch dynamics provides meaningful information about the formation and internalization efficiency of vesicles during clathrin-mediated endocytosis [1,28]. Defects in actin network assembly impact force generation at the membrane and result, in some cases, in ineffective or abortive endocytic events [29–31].

Because effects of mutations can vary in different yeast backgrounds, we aimed at quantifying precisely actin patch dynamics in wild-type (WT) cells and for a variety of mutant cells in the *Saccharomyces cerevisiae* S228C strain used in this study. Selected mutants include single knockouts for genes encoding all the proteins described as actin crosslinkers, namely *sac6Δ*, *scp1Δ*, *abp140Δ*, and the double-mutants *sac6Δ abp140Δ* and *sac6Δ scp1Δ*. We first analyzed precisely the timing and trajectories of actin patches for all strains by recording the fluorescence intensity of the actin-binding protein 1 (Abp1)–green fluorescent protein (GFP) actin reporter over time (Fig 1). We performed this analysis manually over a small number (15–30) of well-defined actin patches (S1A Fig), as well as with an automatic detection method that was less precise but enabled us to analyze phenotypic differences over a much larger number of actin patches (>300). The two methods obtained similar results (S1B Fig). Our results show that actin patches in *abp140Δ* cells have a similar median lifetime to WT cells ($p = 0.6$) (Fig 1B). In agreement with a previous study [12], we found that actin patches in *sac6Δ*, *sac6Δ scp1Δ*, and *sac6Δ abp140Δ* cells have a longer lifetime than WT cells (33%, 33%, and 30% more than WT respectively, $p < 10^{-3}$), but contrary to this study, we found no effect of the deletion of Scp1 on the lifetime of actin patches ($p = 0.81$). Such differences with this previous publication could be explained by the use of different yeast backgrounds between the two studies. Increased patch lifetimes are signatures of a delayed actin network assembly and are correlated with defective internalization of actin patches (Fig 1C and 1D). Analysis of maximum displacements indicates that most patches in WT, *scp1Δ*, and *abp140Δ* cells migrate efficiently up to a median value of 0.3 μm, 0.26 μm ($p = 0.16$), and 0.27 μm ($p = 0.2$), respectively. On the contrary, *sac6Δ* and the double-mutant *sac6Δ scp1Δ* and *sac6Δ abp140Δ* cells have limited movements up to a median value of 0.11 μm, 0.14 μm, and 0.11 μm ($p < 10^{-3}$ as compared to WT) (Fig 1C). The fraction of patches undergoing displacements larger than 200 nm, which is considered as a typical distance above which an endocytic event has successfully occurred, drops from 66% for WT to 33% in *sac6Δ* cells (Fig 1D). Mutants lacking only Scp1 or Abp140 behave

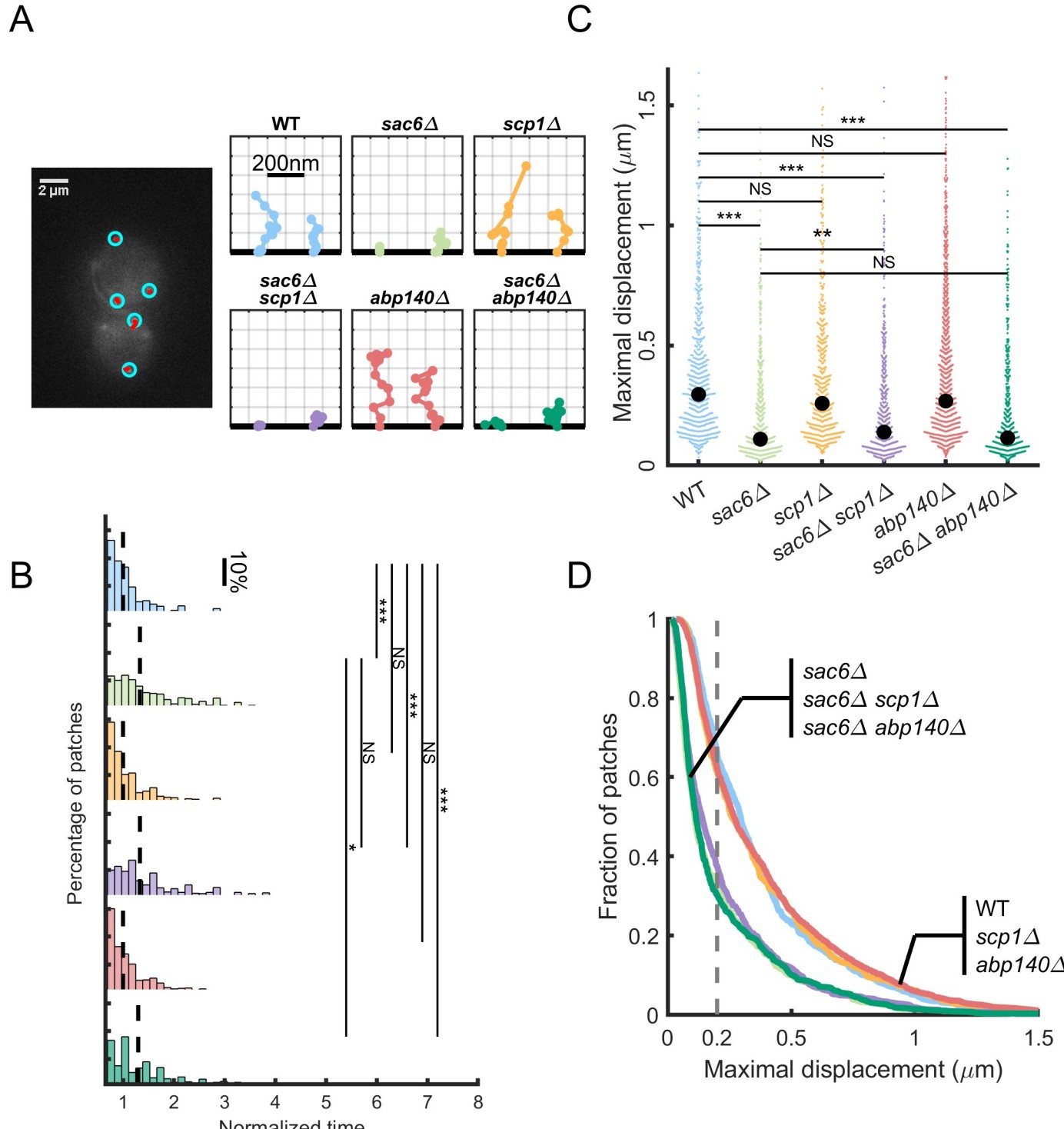

**Fig 1. Endocytic phenotypes in *S. cerevisiae* cells lacking putative crosslinkers.** The underlying data can be found within S1 Data. (A) Observation and tracking of endocytic patches in the different strains used in this study. Left: Epifluorescence image of a representative yeast cell expressing Abp1 fused to GFP (Abp1–GFP) and analyzed to obtain the trajectories of its endocytic actin patches. Actin patches are detected (blue circles) and are progressively tracked (red line) using the plugin TrackMate. Right: Plots showing, for the six different strains analyzed in this study, two typical examples of patch trajectories towards the interior of the cell. The plasma membrane is represented by the horizontal bold line tangent to the *x* axis, while the *y* axis represents the normal to the membrane and is oriented towards the interior of the cell. Identical scales have been used for the two axes. (B) Histogram of actin patch normalized lifetimes extracted from the analysis for the different strains. Lifetimes are normalized by the median of the WT strain for each replicate. Dashed line is the median. (C) Maximal displacement of the actin patches. The maximal displacement represents the distance between the first point of the trajectory and the furthest point inside the cell. Each colored dot

represents a patch. Black dots indicate the median. (D) Endocytosis efficiency. Proportion of patches that have a maximal displacement larger than a given value (*x* axis) for each strain (same color code as in B). The dashed line at 0.2 µm indicates the typical displacement above which endocytic events have successfully occurred. Abp1, actin-binding protein 1; Abp140, actin-binding protein 140; GFP, green fluorescent protein; NS, not significant; Sac6, yeast homolog of fimbrin; Scp1, yeast homolog of calponin; WT, wild type.

similarly to the WT (61% and 62%, respectively), whereas the double mutants *sac6Δ scp1Δ* and *sac6Δ abp140Δ* show an impaired rate of success (37% and 32%, respectively), which is comparable to *sac6Δ* (33%).

Overall, trajectories of actin patches suggest that while Sac6 has a major impact on actin patch internalization and efficient endocytosis, the absence of Scp1 or Abp140 has little to no effect on endocytosis.

## Actin patches of all mutants lacking Sac6 assemble slower and are more numerous

Membrane deformation during endocytosis is powered by the local growth of an actin gel attached to the plasma membrane by adaptor proteins [2]. Defective endocytosis in mutant strains can derive from multiple effects on the actin cytoskeleton. As the level of recruitment of Abp1–GFP varies significantly among strains [10,12], we first aimed at evaluating whether rates of actin assembly in individual endocytic patches were affected in cells.

The patch assembly was monitored in cells by following the fluorescence increase of Abp1–GFP. The rate of actin assembly, which is the slope of the intensity versus time curve during the assembly phase (see Materials and methods), decreases significantly from WT cells to *sac6Δ abp140Δ* and *sac6Δ scp1Δ* cells (68% and 65% of WT rates, respectively; $p < 10^{-4}$), and to *sac6Δ* cells (59% of WT rates, $p < 10^{-4}$) (Fig 2A). The rates of *abp140Δ* cells and *scp1Δ* cells were also lower but to a lesser extent (91% and 87% of WT rates, respectively; $p < 10^{-4}$).

Such effects are unexpected for mutants of proteins usually described as putative crosslinkers and require further investigation to interpret the endocytic phenotypes. We took advantage of a previous protocol in which the formation of actin patches is reconstituted in vitro from yeast cellular extracts around artificial microbeads. Most of the proteins involved in endocytosis, including actin, are soluble and remain present in large amount in these extracts. The networks assembled from the extracts have a similar protein composition to cellular actin patches and include in particular all the crosslinkers mentioned in this study [26]. We investigated whether the slowed-down actin assembly is directly due to the absence of Sac6 by following actin assembly from the extracts in vitro on beads. We grafted superparamagnetic beads of 4.5 µm diameter with the yeast homolog of Wiskott–Aldrich syndrome protein (WASp), Las17, which is a nucleating promoting factor (NPF) of the Arp2/3 complex. These beads are mixed with passivated beads and yeast protein extracts, which triggers the growth of an actin shell from the functionalized beads. During the growth, a low homogenous magnetic field (2 mT) drives the self-organization of the beads into linear chains (Fig 2B). Because of the nanometer resolution on the bead displacements [18], the evolution of the actin shell thickness can be precisely monitored. As previously observed with HeLa cell and *Xenopus* egg protein extracts [32,33], the thickness of the shell reaches a plateau after a few minutes (Fig 2B). We compared the growth of actin networks reconstituted from protein extracts generated from WT, *sac6Δ*, and *scp1Δ sac6Δ* cells (Fig 2B). When reconstituted from a WT protein extract, the growth starts after a delay, and the thickness reaches a plateau within 8 to 10 minutes. In *sac6Δ* protein extracts, actin assembly occurs much faster, and the thickness plateaus within 2 to 3 minutes. This time window is short, and the plateau is usually reached before data acquisition is possible with our experimental setup. The double-mutant protein extract assembles actin

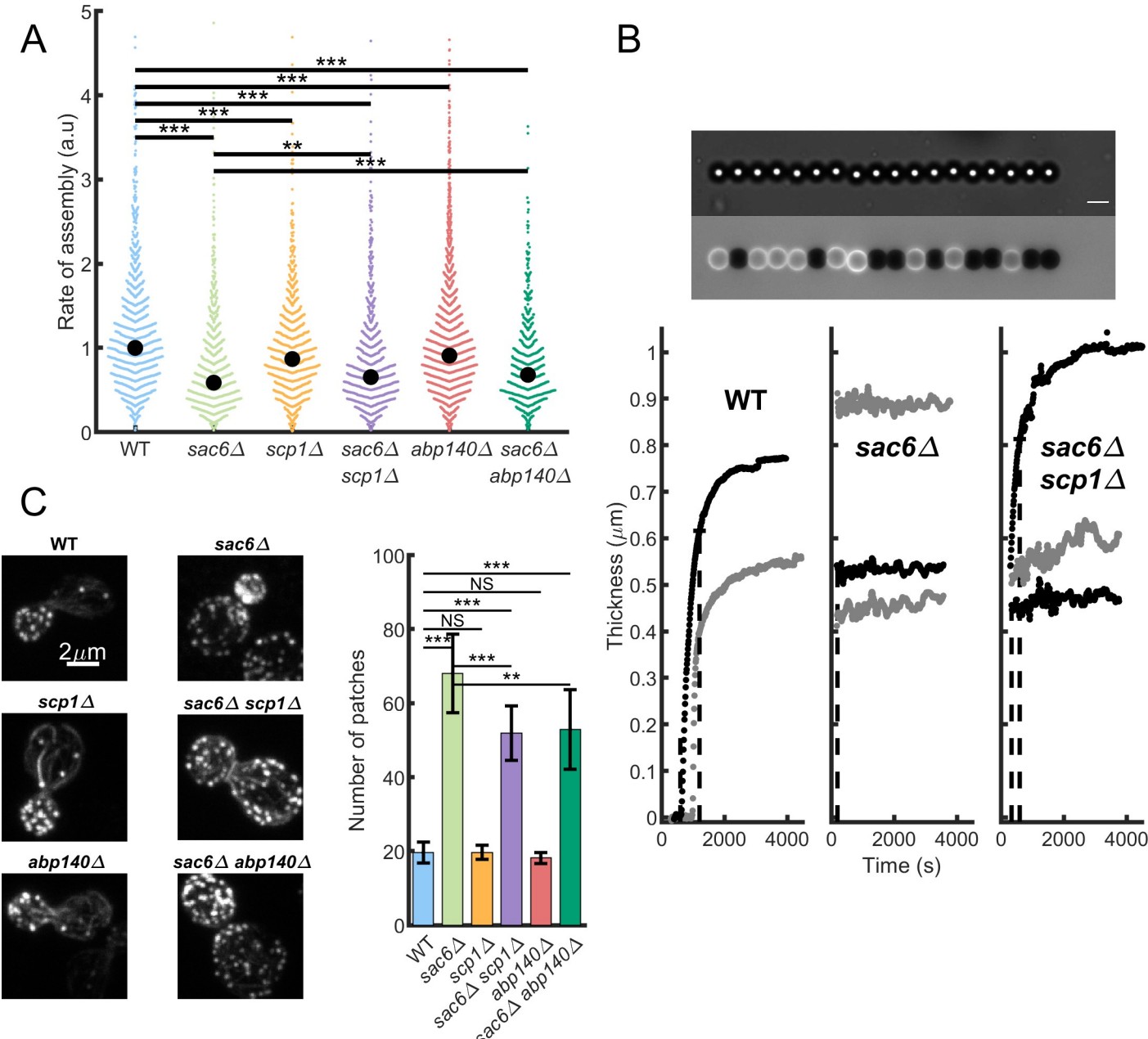

**Fig 2. Dynamics of actin assembly.** The underlying data can be found within S2 Data. (A) Net rate of actin patch assembly in cells. The rates are obtained from the curves of the fluorescence intensity as a function of time during the phase of patch assembly for all strains used in this study. Each colored dot is a measured patch. Black dot is the median. (B) Dynamics of reconstituted actin patch assembly. Growth of actin networks from protein extracts at the surface of magnetic beads. Top: Chain of magnetic beads in the bright field (top) and fluorescence (bottom) channels. On the fluorescence image, actin shells grown around magnetic beads (4.5 μm diameter) are visible in light gray, whereas black beads are nonactivated beads grafted with BSA. The image in bright field is focused below the equatorial plane of the beads to obtain nanometer resolution on the position of the center of the bead. Scale bar is 5 μm. Bottom: Evolution of the thickness of the shell with time for WT, *sac6Δ*, and *sac6Δ scp1Δ* protein extracts. For each strain, a few curves have been represented (gray and black). (C) Number of actin patches for the different strains. Left, typical images used to count the number of patches per cell. Such images are the result of the z-stack projection of the maximum intensities. Actin has been labeled by phalloidin–Alexa568. Right, average number of actin patches in the different strains. Error bars are standard errors. Abp140, actin-binding protein 140; a. u., arbitrary unit; BSA, bovine serum albumin; NS, not significant; Sac6, yeast homolog of fimbrin; Scp1, yeast homolog of calponin; WT, wild type.

shells faster than the WT extract but slower than the *sac6Δ* extract. This qualitative description is confirmed by quantitative measurements of the typical timescale of growth: WT, τ = 830

seconds ± 70 seconds ($N = 27$); *sac6Δ scp1Δ*, τ = 360 seconds ± 40 seconds ($N = 17$); *sac6Δ*, τ = 195 ± 10 seconds ($N = 30$). These results show that actin assembly on the beads is slower in WT extracts than in extracts generated from *sac6Δ* cells. This effect could be explained by a competition between Sac6 binding and Arp2/3 branching observed in vitro on single filaments [34]. The fast growth of actin gels in *sac6Δ* extracts is in opposition to the measured rates of actin assembly in *sac6Δ* cells (Fig 2A) and suggests strongly that the differences observed in the patch assembly dynamics between WT cells and *sac6Δ* cells cannot be attributed to a direct effect of Sac6 on actin nucleation or polymerization.

Another possibility is that the filamentous/globular actin ratio could be altered in some strains, reducing rates of actin assembly through the reduction of the available pool of polymerizable actin. To test this hypothesis, we fixed cells and stained the actin cytoskeleton with fluorescent phalloidin, and the cells were imaged by confocal microscopy (Fig 2C). In agreement with this hypothesis, we found that while the number of actin patches per cell is similar for WT, *abp140Δ*, and *scp1Δ* cells, with, respectively, 19.6 ± 2.8, 19.7 ± 1.9, and 18.2 ± 1.5 patches per cells on average, the number of patches is increased to 68.1 ± 10.6, 51.8 ± 7.4, and 52.9 ± 10.7 patches per cells for *sac6Δ*, *sac6Δ scp1Δ*, and *sac6Δ abp140Δ* cells, respectively (mean ± standard error). Our results show that the slower patch assembly observed in mutants lacking Sac6 is possibly due to an excessive assembly of actin in cells rather than to a direct effect of Sac6 on the actin polymerization dynamics. An increasing number of patches with a constant pool of actin should lead to a reduced amount of monomeric actin available and a slowed-down actin patch assembly.

## Networks reconstituted from yeast protein extracts are mechanically stiffer than those reconstituted from a minimal mix of purified yeast proteins

We investigated next the mechanical properties of actin patches reconstituted from the different extracts, expecting that the absence of crosslinkers may have a quantifiable impact on the elastic properties of the gels. These measurements were performed with the same superparamagnetic beads as the ones used in the previous section. After growth at low magnetic field (2 mT) that drives the self-organization of the beads into linear chains (Fig 2A), the magnetic field is ramped up to 80 mT for 7.5 seconds. This timescale has been chosen to match the typical timescale of endocytosis measured in vivo (see previous section). Increasing the magnetic field increases the dipolar magnetic force between each pair of microbeads from a few piconewtons to about one nanonewton. This force deforms the shell of actin present on one of the two beads, and the shell thickness is measured by video microscopy during deformation (Fig 3A). We analyzed the deformation of the shell as a function of the imposed force to get the elastic properties of the shell. The procedure includes fitting the force-deformation curve with a Hertz model of the contact between elastic spheres, modified to take into account the limited thickness of the elastic shell [35].

Actin networks reconstituted from WT yeast extracts have an elastic modulus of median value of 5.7 kPa ($n = 107$). The distribution of measured values is rather broad and is best represented by a logarithmic distribution (Fig 3A). The broad distribution validates our experimental approach, the throughput of which, as compared to other techniques, is relatively high because of the self-organization of the beads [18].

We compared networks reconstituted from yeast protein extracts to networks reconstituted from a minimal mix of purified yeast proteins. This minimal mix is composed of actin, profilin, Arp2/3 complex, and capping protein [36–38]. We used concentrations of actin, Arp2/3, and capping protein that were measured in the extracts by western blot [39]. Profilin was added at a 3-fold excess over actin. Magnetic bead experiments showed that these networks

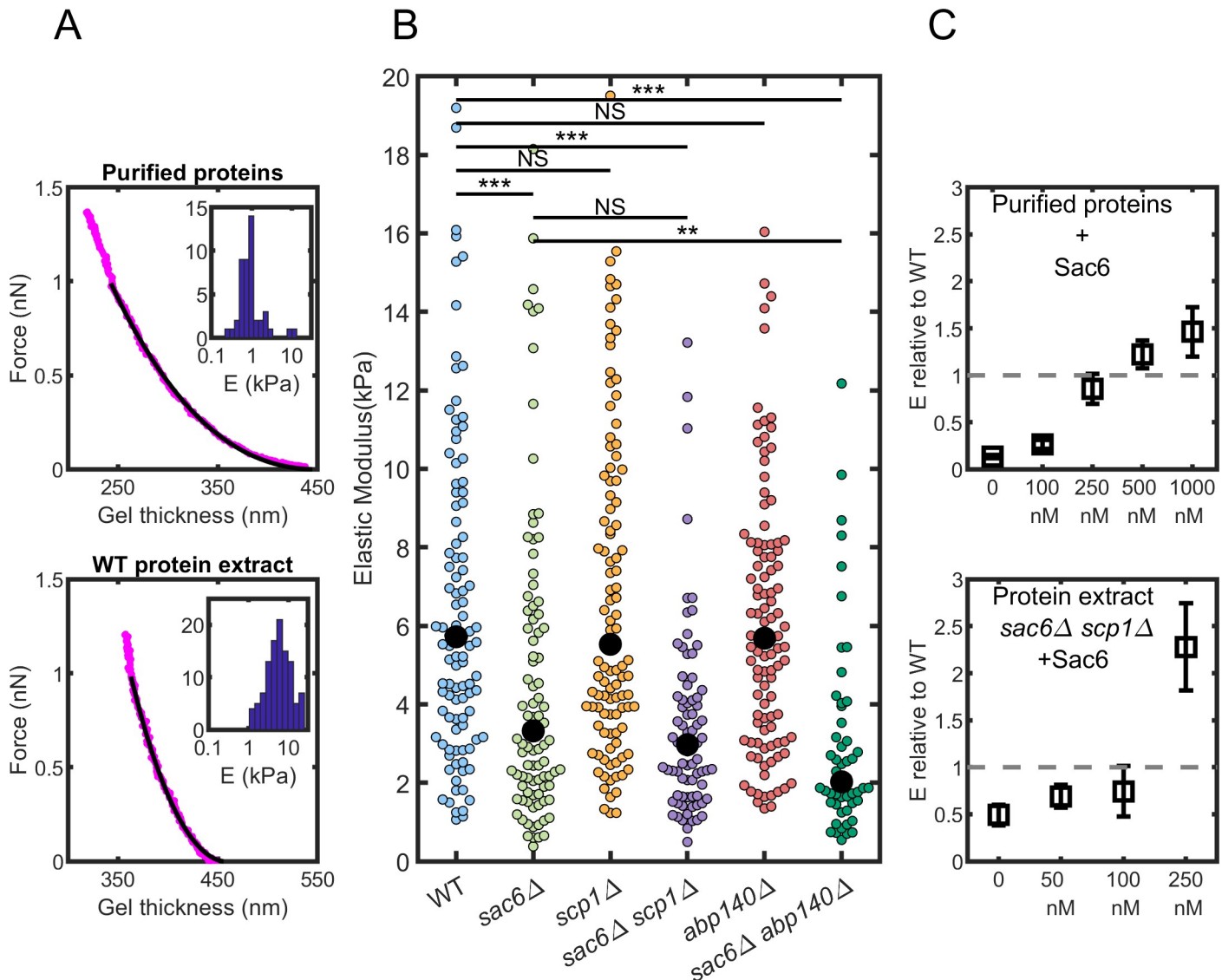

**Fig 3. Elastic properties of reconstituted actin networks.** The underlying data can be found within S3 Data. (A) Principle of the measure. Force-distance curves obtained on an actin network and distribution of the elastic moduli extracted from many force-distance curves (inset). Gels are assembled from a mix of purified proteins (top, 46 gels) or from WT protein extracts (bottom, 107 gels). (B) Elastic moduli of the actin gels assembled from protein extracts. Colored symbols are individual measurements; black dots are the median values. (C) Effect of the addition of purified Sac6 on the elastic modulus. Elastic moduli normalized by the elastic modulus of WT protein extract when Sac6 is added to mix of purified proteins (top) or *sac6Δ scp1Δ* protein extract (bottom). Error bar calculation is described in the Materials and methods section. Abp140, actin-binding protein 140; NS, not significant; Sac6, yeast homolog of fimbrin; Scp1, yeast homolog of calponin; WT, wild type.

were much softer (Fig 3A). The median elastic modulus was measured at 0.86 kPa ($n = 46$), which is widely different from the elastic modulus of gels assembled in yeast extracts ($p < 10^{-4}$). This difference can be due to the absence of structural proteins such as crosslinkers or to a change in the network architecture due to other accessory proteins present in the extract. This value is also significantly smaller than previously published measurements on actin networks grown in a similar system of purified mammalian proteins [17,27]. This is, however, not very surprising because actin filaments purified from budding yeast are known to be more flexible than actin filaments purified from skeletal muscles [40].

## Sac6, Scp1, and Abp140 impact actin patch stiffness differently

To investigate the effect of Sac6, Scp1, and Abp140 on the elasticity of the actin networks, we reconstituted networks with protein extracts from mutant knockouts (Fig 3B). The elastic modulus for actin networks reconstituted in *sac6Δ* extracts is significantly softer than for networks assembled in WT extracts, with a median at 3.3 kPa ($p < 10^{-4}$, $n = 98$). On the contrary, our measurements show no significant softening for actin networks reconstituted in *scp1Δ* and *abp140Δ* extracts, with respective elastic modulus medians of E = 5.5 kPa ($n = 109$) and E = 5.7 kPa ($n = 106$). To test whether the presence of Sac6 could mask an effect on the elasticity from Scp1 and Abp140, we also tested extracts from double knockout mutants lacking Sac6 and either Scp1 or Abp140. Actin networks reconstituted from the double knockout mutant *sac6Δ scp1Δ* were measured at a median modulus E = 3.0 kPa ($n = 79$), which is softer than for WT extracts ($p < 10^{-4}$) but not significantly different from *sac6Δ* extracts ($p < 0.2$). On the contrary, actin networks assembled from the double knockout *sac6Δ abp140Δ* extracts were measured at a median modulus E = 2.0 kPa ($n = 51$), which is softer than both WT ($p < 10^{-4}$) and *sac6Δ* extracts ($p < 0.005$).

In conclusion, the absence of Sac6 has an effect on the elasticity in every combination tested here (*sac6Δ*, *sac6Δ scp1Δ*, and *sac6Δ abp140Δ*). The absence of Scp1 does not change the elasticity of reconstituted networks even in the absence of Sac6, whereas the absence of Abp140 softens the networks, but only in the absence of Sac6.

## Addition of purified Sac6 restores the rigidity of reconstituted actin patches

We wanted to control whether the softening of actin gels was really due to an absence of the proteins from the meshwork and not to any other indirect effect during the assembly. To verify the central role of Sac6 in the rigidity of actin networks, we purified Sac6 and added it to both biomimetic assays, i.e., when actin gels are assembled from a minimal mix of protein or from the *sac6Δ scp1Δ* cell extract (Fig 3C). In both cases, the elastic modulus increases as a function of the Sac6 concentration in a dose-dependent manner.

On actin gels assembled from cell extracts, the elastic modulus is not significantly different from actin gels assembled in WT extracts when *sac6Δ scp1Δ* extracts are supplemented with 100 nM Sac6 prior to the experiments. They become significantly more rigid at 250 nM ($p < 10^{-6}$). On actin gels assembled with purified proteins, the elastic modulus reaches the value of WT extracts actin gels when [Sac6] = 250 nM, which is a value larger than the value measured in the extracts (25 nM). Above 250 nM of Sac6, the elastic modulus overpasses the value of the WT (Fig 3C). The fact that the amount of Sac6 that needs to be added to the mix of purified proteins is larger than the value measured in the extracts highlights the role of the other accessory proteins present in the cell extract. These proteins may impact the architecture of the networks; among them, one can think of unidentified crosslinkers or other side-binding proteins that will modify the flexibility of the filaments.

## Discussion

### Sac6 plays a dominant role in actin networks rigidity

In the past, most studies have investigated the impact of actin accessory proteins on the mechanical properties of actin networks separately from their phenotypic effect in cells. The experimental setup presented in this study allows for the closest comparison possible between the effect of actin crosslinkers for clathrin-mediated endocytosis and their quantitative contribution to actin network stiffening.

Our results demonstrate that actin crosslinking stiffens actin networks at endocytic sites. Our results also reveal that the three putative actin crosslinkers stiffen actin networks with different efficiencies. Sac6 plays a major role in the stiffening of actin patches, and its absence is sufficient to decrease dramatically the rigidity of actin networks by 42%. Abp140 also seems to contribute to the stiffening of actin networks, although to a lesser extent: its absence in the presence of Sac6 does not soften actin patches, but its absence combined with an absence of Sac6 has a clear softening effect on actin networks (65%). A possible explanation could be that Sac6 dominates the stiffening of actin networks so that any contribution of Scp1 or Abp140 is not detectable in its presence. Another explanation could be that Abp140 and Sac6 compete to a certain extent in their binding to actin, for example, if Sac6 binds to actin filaments with a higher affinity than Abp140. Abp140 may then act as a mechanical rescue that would compensate for an absence of Sac6. The situation is very different for Scp1, for which we could not detect any significant reduction of actin network stiffness in its absence even in the absence of Sac6. These observations do not necessarily prove that Scp1 has no crosslinking activity but that, in a physiological context, its effect is negligible when compared to Sac6. It is possible that the affinity of Scp1 for actin filaments is lower than that of Abp140 and Sac6. Alternatively, Scp1 could have a typical timescale of binding and unbinding much smaller than the timescale probed here (7.5 seconds), which was chosen to be as close as possible to the duration of endocytosis. Scp1 may attach and detach many times during the lifetime of the patch, resulting in the absence of rigidification of the actin network both in vivo and in vitro.

## Relation between patch rigidity and endocytosis efficiency

In yeast endocytosis, actin networks contribute to the invagination of the plasma membrane, and the force exerted by the polymerization of the actin filaments is needed to counteract the turgor pressure. In current mechanistic models, for this force to be effective, it is postulated that the network is attached to the deformed membrane via the coat proteins and polymerizes from the plasma membrane, pushing the network and the invaginated membrane inside the cell [41,42]. The turgor pressure acts as a force by unit surface that opposes the growth of the actin network. Because the network is elastic, the opposing force can compress the growing network, limiting its lengthening. If the network developing the force is too soft, most of the growth will be counteracted by this compression. In the following, we outline a simple model to support this idea. We consider an elastic material that gains a slab of thickness l in a time $t$ because of the growth process and that is submitted to a compressive force by unit area $-\sigma_{op}$. Because the new slab is deformed, the actual increase in thickness ($x$) is going to be smaller than l. In a linear elastic material, the compressive force and the strain are related by Hooke's law: $\sigma = E\,\varepsilon$, where $\sigma$ is the force by unit area (called stress in mechanics), E the Young modulus of the material, and $\varepsilon = \log(1 + \delta l/l)$ the true strain experienced by the material with $\delta l = -(l - x)$. From the previous expression, we get $-\sigma_{op} = E\log(x/l)$. Hence the increase in thickness of the growing gel is $x = l\exp(-\sigma_{op}/E)$. In term of gel growth speed, this increase leads to $v = l/t\,\exp(-\sigma_{op}/E)$ by dividing the previous equation by $t$.

If the opposing force by unit area is small compared to the elastic modulus ($\sigma_{op} \ll E$), the deformation is limited, and the effect of the gel elasticity is minimal on the growth speed. One can simplify the equation to $v = l/t(1 - \sigma_{op}/E)$. However, if the opposing stress is of the same magnitude as the elastic modulus ($\sigma_{op}$ approximately E), the compression in the gel is important, and the speed of growth will depend exponentially on the elasticity of the network.

In this paper, we have measured the elastic modulus of an actin network resembling the actin structure assembled during yeast endocytosis. We obtained values of 5.7 kPa for gels assembled from WT extracts and 3.3 kPa for gels assembled from *sac6Δ* extracts. These moduli

can be compared to the force by unit area opposing the growth. In the case of endocytosis in yeast, the opposing force is mainly due to the turgor pressure that hampers invagination of the membrane inside the yeast. Pressure as high as 0.6 MPa has been reported [43]. If one would directly take the turgor pressure as the opposing stress, an unrealistic large deformation would be obtained: the actin network is too soft to grow under such pressure. A few factors can, however, limit the magnitude of the opposing stress. If the surface on which the turgor pressure is acting, i.e., the patch of clathrin that will form the vesicle, is smaller than the surface of the actin network, the stress opposing the growth will be reduced by the ratio of these areas. This seems to be the case, as recent super-resolution imaging of endocytic patches measured the coat proteins extending to 20–50 nm outer radius, while the actin-associated proteins extended to 80–100 nm [5]. Another factor that has been proposed to mitigate the opposing stress is the lever effect [44]: if the actin filaments are pushing at an angle on the coat proteins, the force opposing the growth will be lowered. The tight organization of the endocytic patch with actin activators at the periphery of the patch [5] would also facilitate this effect. Although a precise estimation of the opposing force in endocytosis is not within our grasp yet, it is reasonable to assume this stress is of the same order of magnitude as the typical elastic moduli of actin networks. The elasticity of the network is thus going to heavily impact its growth speed.

## Consequences of defective actin crosslinking on clathrin-mediated endocytosis

The precise characterization of the contribution of actin crosslinkers to actin network rigidity enabled us to investigate possible correlations with the corresponding phenotypes in yeast cells. We analyzed the effects of an absence of crosslinkers at the patch scale and at the whole cell scale.

We observed that loss of actin patch stiffness is well-correlated with a decreased efficiency of endocytosis. This is quantified by the increased patch lifetime and the reduced internalization efficiency, which are both clear signatures of defective endocytosis. In addition, the rate of actin assembly is also lower in mutant cells. Therefore, defective endocytosis could be the result of softer actin networks, slower rates of actin assembly, or both. Increased patch lifetime is linked to the presence of a larger number of actin patches in mutant cells, and it is therefore likely that a larger fraction of actin is polymerized in these cells. It is now well-accepted in the field that the pool of polymerizable actin is not in large excess but is, on the contrary, limited to the point that the different actin networks assembled in the cell compete for this limited reservoir [45,46]. In this context, it is likely that a larger number of patches in the mutant cells correspond to a larger amount of polymerized actin and therefore to a lower amount left of polymerizable actin.

## Conclusion

Overall, we suggest that our data are consistent with a chain-of-consequences model (Fig 4). In this model, we propose that the first effect of a decrease in actin crosslinking is to impact the stiffness of actin networks. As a consequence, actin networks are less efficient in transducing forces to the membrane, and failed endocytosis increases actin patch lifetime. A longer lifetime of actin patches for an equivalent rate of endocytic events initiation explains why more actin patches are present in cells on average. As actin networks compete for a limited pool of actin monomers, an increase in the number of actin patches increases the amount of polymerized actin and decreases the pool of monomeric actin. Lower amounts of monomeric actin therefore reduce the rate of actin assembly, which causes further failures of endocytosis.

## WT

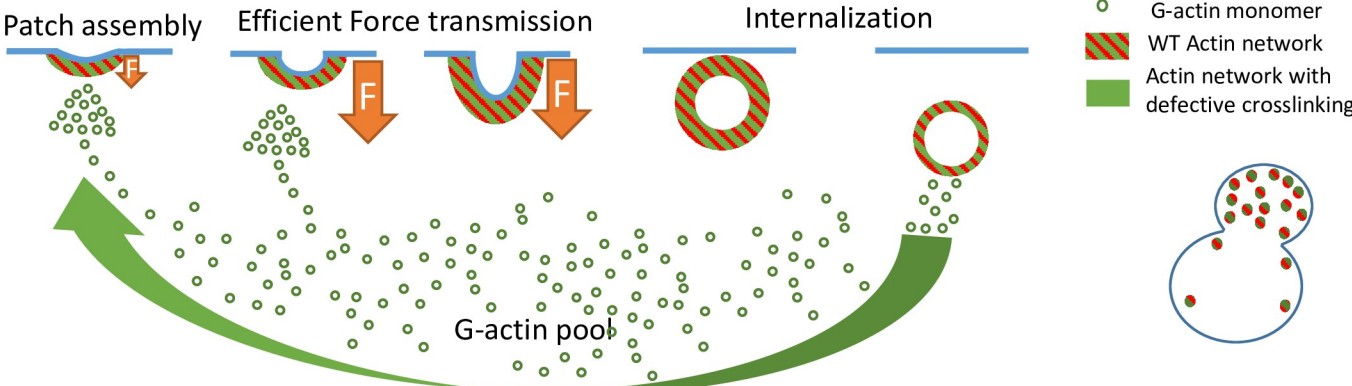

## *Defective actin crosslinking*

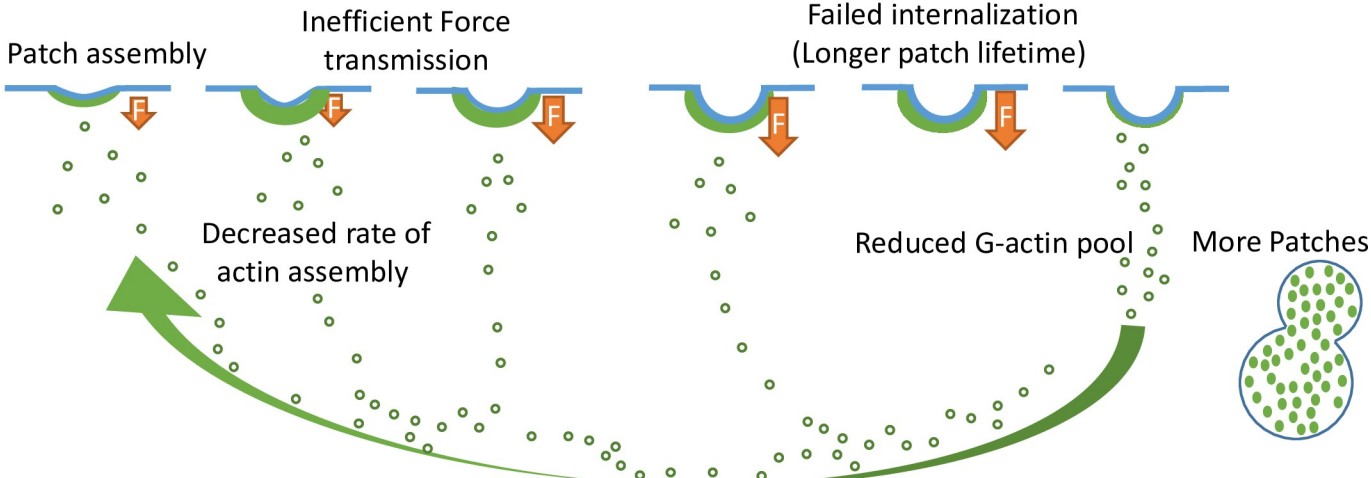

**Fig 4. Schematic of a chain-of-consequence model for yeast endocytosis.** This model proposes that in WT cells (upper panel), the mechanical properties of actin patches (green filled dots) are optimized for efficient force transmission at the plasma membrane (in blue) and a rapid internalization of the endocytic vesicles. Subsequently to the internalization of an endocytic vesicle, the actin network disassembles and contributes to replenishing the limiting pool of G-actin monomers (green open dots) up to a normal level. In mutant strains, defective actin crosslinking (crosslinkers are in red) impacts the stiffness of actin networks. Force transduction to the membrane is less efficient, increasing actin patch lifetime and the proportion of failed endocytic events. Longer lifetime of actin patches for an equivalent frequency of endocytic events initiation accounts for a higher number of actin patches in these mutant cells. Because the pool of actin monomers is limiting, an increased number of actin patches means that more actin is polymerized in cells and that less monomeric actin is present. Depletion of the pool of monomeric actin leads to a reduced rate of actin assembly, which causes further failures of endocytosis. WT, wild type.

## Materials and methods

### Yeast strains

Yeast mutants were generated with standard procedures as described in [47] and are derived from a S228C strain.

## Yeast cell imaging and analysis

**Live cell imaging.**   Cells were cultured at OD 0.5 to 0.7 and immobilized on coverslips coated with $0.1 \text{ mg} \cdot \text{ml}^{-1}$ of concanavalin A. Cells were imaged using a Zeiss Axio Observer Z1 microscope equipped with a 100×/1.4 NA oil Ph3 Plan-Apochromat objective (Zeiss, Oberkochen, Germany) and a Hamamatsu ORCA-Flash 4.0 LT camera (Hamamatsu, Hamamatsu City, Japan). Images were acquired with Zen 2.3 blue edition software. Endocytic actin patch tracking analysis was performed by following the centroid intensity of Abp1-GFP patches using the Fiji plugin TrackMate [48] (https://imagej.net/TrackMate). Individual patches were detected with a median filter and a difference of Gaussian (DoG) particle detection algorithm for spots with a diameter of approximately 0.5 μm. A selection of particles was subsequently done for particles with a contrast above 0.1. With the Linear Assignment Problem tracker, the patch tracks were reconstituted. From the tracks files, we plotted the lifetime and the maximum displacement, which is the distance between the initial and the furthest position on the trajectory. The data were processed through custom scripts in MATLAB R2017b (The MathWorks, Natick, MA, USA). For each patch, position, lifetime, maximum displacement, and intensity were recorded for the analysis, and only patches with a lifetime longer than 5 seconds and displaying an increase followed by a decrease in intensity were kept for the analysis [49]. From the intensity over time curve, we measured the slope of the intensity during the actin assembly (from 4 seconds before the maximum of the intensity). Because the patch lifetime and the rate of actin assembly significantly differed between replicates, we normalized them by the median of the WT observed on the same day, pooled the normalized data of the different strains, and tested the statistical significance of the pooled data with Wilcoxon rank test (Fig 1B).

**Phalloidin staining.**   Cells were cultured at 25˚C in standard rich media (YPD) and collected at OD 0.5–1, fixed in 4% formaldehyde/YPD for 2 hours, washed twice in PBS, and stained overnight with 250 nM phalloidin-Alexa568 (Invitrogen Ref. A12380; Carlsbad, CA, USA). Cells were imaged using a Leica TCS SP8 X White Light Laser confocal microscope (Leica, Wetzlar, Germany) equipped with an HC PL APO CS2 100×/1.4 NA oil objective and a hybrid detector. Z-stack images were collected every 0.3 μm with Las X 3.5.5.19976 software. Patches were counted manually from the maximum intensity z-stack projections.

## Yeast extract preparation

Yeast extracts were prepared based on a published protocol [50]. Briefly, yeast strains were cultured in standard rich media (YPD) at 30˚C to an $OD_{600}$ of 1.5–2. Cells were harvested by centrifugation, resuspended in cold water, and centrifuged again. Pellets were flash-frozen in liquid $N_2$ and ground by mechanical shearing in a Waring blender. To each gram of yeast powder were added 100 μl of Hepes buffer (100 mM [pH 7.5]) and 10 μl of protease inhibitors (Protease Inhibitor Cocktail Set IV, Calbiochem, Merck4Biosciences). Yeast powder was gently mixed on ice with the buffer, progressively thawed, and centrifuged for 20 minutes at $50,000 \times g$. The cleared supernatant was collected, kept on ice, and used within 3 hours.

## Protein expression, purification, and labeling

**Yeast actin purification and labeling.**   *S. cerevisiae* actin was purified from commercially purchased baker's yeast (Kastalia, Lesaffre, Marcq-en-Baroeul, France) as described in [51] and labeled with Alexa dyes as described in [52].

**Yeast WASp (Las17), profilin (Pfy1) and capping protein (Cap1 and Cap2).**   *S. cerevisiae* WASp, profilin, and capping protein were expressed and purified from *Escherichia coli* Rosetta 2(DE3)pLysS cells as described in [53].

**Yeast Arp2/3 complex.** Endogenous *S. cerevisiae Arp2/3* complex was purified from a myc-tagged yeast strain as described in [54].

**Yeast fimbrin (Sac6).** *S. cerevisiae*'s fimbrin was overexpressed from a multicopy plasmid (2 μ URA3 *Pgal1-SAC6-9×HIS*) in yeast (*MATa*, *leu2*, *ura3-52*, *trp1*, *prb1-1122*, *pep4-3*, *pre1-451*) under the control of a GAL1 promoter. Protein expression was induced for 12 hours at 30°C with 2% galactose. Cells were harvested by centrifugation, frozen in liquid nitrogen and grinded in a steel blender (Waring, Winsted, CT, USA). For protein purification, 5 g of ground yeast powder was mixed with 45 ml of HKI10 buffer (20 mM Hepes [pH 7.5], 200 mM KCl, 10 mM imidazole [pH 7.5]) supplemented with 50 μl of protease inhibitors (Set IV, Calbiochem, Merck4Biosciences, Darmstadt, Germany), and thawed on ice. The mixture was centrifuged at $370,000 \times g$ for 40 minutes, and the supernatant was incubated with 500 μl bed volume of Ni-Sepharose 6 Fast Flow (GE Healthcare Life Sciences, Piscataway, NJ, USA) for 2 hours at 4°C. Bound protein was batch purified with HKI500 buffer (20 mM Hepes [pH 7.5], 200 mM KCl, 500 mM imidazole [pH 7.5]), concentrated with an Amicon Ultra 4 ml device (Merck4Biosciences), dialyzed for 2 hours in HKG buffer (20 mM Hepes [pH 7.5], 200 mM KCl, 6% glycerol), aliquoted, and flash-frozen in liquid nitrogen.

## Actin assembly on superparamagnetic beads

**Functionalization and passivation of beads.** Superparamagnetic microspheres (Dynabeads M-450 Epoxy, 4.5 μm diameter; Thermo Fisher Scientific, Waltham, MA, USA) were diluted 10 times in HK buffer (20 mM Hepes [pH 7.5], 150 mM KCl) and incubated with 100 nM Las17 for 30 minutes on ice. Tubes were rotated during incubation to avoid the sedimentation of the beads. Beads were then saturated with 1% BSA for 15 minutes, washed, and eventually stored on ice in HK buffer supplemented with 0.1% BSA.

**Actin assembly from yeast extracts.** 0.75 μl of Las17-functionalized microbeads and 0.75 μl of BSA-passivated microbeads were added to 28.5 μl of yeast extract supplemented with 0.1 to 0.3 μM of Alexa568-labeled actin (10%–5% labeled) to induce formation of branched actin networks. Actin networks were assembled for 20–30 minutes before introduction into homemade flow chambers. Chambers were sealed with a mix of 1/3 Vaseline, 1/3 lanolin, and 1/3 paraffin.

**Actin assembly from purified proteins.** 0.75 μl of Las17-functionalized microbeads and 0.75 μl of BSA-passivated microbeads were added to a protein mix containing 1.2 μM Alexa568-labeled actin, 3.6 μM profilin, 25 nM Arp2/3 complex, and 400 nM capping protein in a motility buffer containing 20 mM Hepes (pH 7.5), 100 mM KCl, 2 mM EGTA, 2 mM MgCl$_2$, 3 mM ATP, 5 mM DTT, and 1.5% BSA.

## Timescale of actin network growth around beads

For each pair of beads, we measured the gel thickness at the plateau by averaging the thickness over the last 500 seconds. For gels assembled from WT protein extracts, the time $t_0$ at which the growth starts and the time $t_{80}$ at which the thickness is 80% of the final thickness are measured from the curves. The timescale of the growth τ is the difference between $t_{80}$ and $t_0$. When gels were assembled from *sac6Δ* extracts, the plateau was reached before the recording of the thickness was possible: τ was then considered as the first time point, even if this is an overestimation as compared to WT. This was also the case for some gels assembled from the double-mutant protein extracts. For these gels, when possible, $t_{80}$ was measured, and τ is directly chosen to be $t_{80}$ because $t_0$ is not measurable.

## Mechanical measurements

The mechanical experimental setup consists of a Zeiss Axio A1 inverted microscope with a modified stage hosting two coils of diameter 88 mm and width 44 μm, with a soft iron core

and 750 turns of copper wire. A Bipolar Operational Power supply (Kepco, New York City, NY, USA) feeds the coils with up to 5 A electrical current corresponding to an 80 mT homogenous magnetic field just above the objective. The objective is a 100× oil immersion Apochromat with 1.4 NA. Timelapse images were recorded with an ORCA-Flash 4.0 CMOS camera (Hamamatsu). A Labview (National Instruments, Austin, TX, USA) custom program allows controlling of the field with simultaneous image acquisition.

All chains of beads present in the experimental chamber undergo deformation during the ramp up of the field. To avoid any plastic effect that would deform permanently the zone of contact between the beads, we used a new experimental chamber with fresh actin shells for each repetition.

Data analysis was performed with Image J (NIH, Bethesda, MD, USA) and MATLAB. The position of the bead center was determined with the center of mass of the white pixels on images taken at a focus below the equatorial plane of the beads. The force was calculated from the value of the field, the magnetic susceptibility of the beads, and their position. The value of the elastic modulus was obtained with a fit of the deformation of a shell by a bead as a function of the force. A criterion of goodness of fit ($R^2 > 0.9$) was used to remove aberrant curves. The comparison between the networks reconstituted from different mutant was made with a Wilcoxon–Mann–Whitney nonparametric test. For the experiments with an increasing concentration of purified crosslinkers, a random effect model from meta-analysis [55] was used to take into account a higher variability of measure between different extract preparations than within the same extract preparation. This model uses the inverse variance inside a sample as a weight for the calculation of the mean. Standard errors were computed from both these weights and the variance between samples.

## Supporting information

**S1 Fig. Comparing the TrackMate method with a manual tracking of the patches.** (A) Example of the intensity of an actin patch over time. The manual and automatic tracking lifetimes are indicated with a black arrow and gray arrow, respectively. The higher threshold on detection explains the shorter lifetime measured with TrackMate. The slope of the red dashed line represents the rate of actin assembly. (B) Manual tracking of patch lifetime in the different strains. Each colored dot represents a patch. Black dots indicate the median. (C) Patch lifetime distribution. Patches with a lifetime lower than 5 seconds (dashed line) were not taken into account to avoid detection artifacts.
(TIF)

**S2 Fig.** (A) Histogram of patch lifetime extracted from the analysis for the different strains. The median (represented by a dashed line) is 8.0 seconds for WT, 7.2 seconds for *scp1Δ* and *abp140Δ*, 10.4 seconds for *sac6Δ* and *sac6Δ abp140Δ*, and 9.6 seconds for *sac6Δ scp1Δ*. The data are pooled from several replicates and non-normalized. (B) Rate of actin patch assembly in the cells. The rates are obtained from the curves of the fluorescence intensity as a function of time during the phase of patch assembly for all strains used in this study. Each colored dot is a measured patch. The median (represented by a black dot) is 1,800 a.u. for WT, 2,100 a.u. for *scp1Δ*, 1,900 a.u. for *abp140Δ*, 1,200 a.u. for *sac6Δ*, 1,000 a.u. for *sac6Δ abp140Δ*, and 1,600 a.u. for *sac6Δ scp1Δ*. The data are pooled from several replicates and non-normalized. Abp140, actin-binding protein 140; a.u., arbitrary unit; Sac6, yeast homolog of fimbrin; Scp1, yeast homolog of calponin; WT, wild type.
(TIF)

**S1 Data. Raw data for Fig 1.**
(XLSX)

**S2 Data. Raw data for Fig 2.**
(XLSX)

**S3 Data. Raw data for Fig 3.**
(XLSX)

## Acknowledgments

The authors thank Laurent Blanchoin and Christopher P. Toret for fruitful discussions. RB, OdR, and JH are members of the French GDR 3070 (GDR Celltiss).

## Author Contributions

**Conceptualization:** Olivia du Roure, Alphée Michelot, Julien Heuvingh.

**Data curation:** Jessica Planade, Reda Belbahri.

**Formal analysis:** Jessica Planade, Reda Belbahri, Julien Heuvingh.

**Funding acquisition:** Olivia du Roure, Alphée Michelot, Julien Heuvingh.

**Investigation:** Jessica Planade, Reda Belbahri, Micaela Boiero Sanders, Audrey Guillotin.

**Methodology:** Olivia du Roure, Alphée Michelot, Julien Heuvingh.

**Project administration:** Olivia du Roure, Alphée Michelot, Julien Heuvingh.

**Resources:** Olivia du Roure, Alphée Michelot, Julien Heuvingh.

**Supervision:** Olivia du Roure, Alphée Michelot, Julien Heuvingh.

**Validation:** Olivia du Roure, Julien Heuvingh.

**Visualization:** Reda Belbahri, Olivia du Roure, Julien Heuvingh.

**Writing – original draft:** Reda Belbahri, Olivia du Roure, Alphée Michelot, Julien Heuvingh.

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
