## [Editor Report · Decision Letter 0]

9 Aug 2019

Dear Dr Michelot, 

Thank you for submitting your manuscript entitled "Mechanical stiffness of reconstituted actin patches correlates tightly with endocytosis efficiency" for consideration as a Short Reports by PLOS Biology.

Your manuscript has now been evaluated by the PLOS Biology editorial staff as well as by an academic editor with relevant expertise and I am writing to let you know that we would like to send your submission out for external peer review.

*Please be aware that, due to the voluntary nature of our reviewers and academic editors, manuscripts may be subject to delays during the holiday season. Thank you for your patience.*

Please re-submit your manuscript within two working days, i.e. by Aug 11 2019 11:59PM.

Kind regards,

Di Jiang, PhD

Associate Editor

PLOS Biology

---

## [Decision Letter · Decision Letter 1]

11 Sep 2019

Dear Dr Michelot,

Thank you very much for submitting your manuscript "Mechanical stiffness of reconstituted actin patches correlates tightly with endocytosis efficiency" for consideration as a Short Reports by PLOS Biology. Your paper was evaluated by the PLOS Biology editors as well as by an Academic Editor with relevant expertise and by two reviewers. 

Based on the reviews, we will probably accept this manuscript for publication, providing that you will modify the manuscript according to the review recommendations.

We expect to receive your revised manuscript within two weeks. Your revisions should address the specific points made by each reviewer. In addition to the remaining revisions and before we will be able to formally accept your manuscript and consider it "in press", we also need to ensure that your article conforms to our guidelines. One requirement is described below under 'DATA POLICY' and is marked with '***IMPORTANT: '. A member of our team will be in touch shortly with a set of requests. As we can't proceed until these requirements are met, your swift response will help prevent delays to publication.

Please note that you may have the opportunity to make the peer review history publicly available. The record will include editor decision letters (with reviews) and your responses to reviewer comments. If eligible, we will contact you to opt in or out.

Sincerely,

Di Jiang, PhD

Associate Editor

PLOS Biology

DATA POLICY:

***IMPORTANT: Regardless of the method selected, please ensure that you provide the individual numerical values that underlie the summary data displayed in the following figure panels: 1A-D, 2A-C, 3A-C, S1A-C, as they are essential for readers to assess your analysis and to reproduce it. ***IMPORTANT: Please also ensure that figure legends in your manuscript include information on where the underlying data can be found. You can write, for example: 'Values for each data point can be found in S1 Data.'.

For manuscripts submitted on or after 1st July 2019, we require the original, uncropped and minimally adjusted images supporting all blot and gel results reported in an article's figures or Supporting Information files. We will require these files before a manuscript can be accepted so please prepare them now, if you have not already uploaded them. Please carefully read our guidelines for how to prepare and upload this data: https://journals.plos.org/plosbiology/s/figures#loc-blot-and-gel-reporting-requirements.

Reviewer remarks:

Reviewer #1: see attached file

Reviewer #2: This manuscript addresses the role of mechanical stiffness of the endocytic actin network in the budding of endocytic vesicles in yeast cells. The mechanical properties of the endocytic protein machinery are a very poorly studied due to significant technical challenges in analyzing these tiny cellular structures. This question, however, is critical for mechanistic understanding of endocytosis, and for understanding other mechanical processes in cells, such as cell migration. The authors use a new and very innovative approach to tackle this question. They use in parallel an in vitro approach and an in vivo approach, and correlate the results from these two complementary approaches. The in vitro assay allows them to directly measure the mechanical stiffness of reconstituted endocytic actin patches with or without key actin filament crosslinkers. The in vivo approach reveals the endocytic functionality of the wildtype and comparable crosslinker mutant strains. The manuscript shows that crosslinker proteins increase the stiffness of the actin patch and that enough stiffness is essential for the proper budding of endocytic vesicles. The results and the approach are very interesting for the field of endocytosis, but also for the broader audience of people working on the cytoskeleton and cellular biomechanics. 

The manuscript is very clear and concise. The experiments are well performed and controlled. I have only very minor suggestions for the authors. 

- In the abstract use “actin crosslinkers” instead of just “crosslinkers”.

- Fig 1B: label the strains. There are no “black dots” in the fig 1B as stated in the legend. 

- Fig 4: Although the model figure is quite clear, it would be helpful to have a figure legend that briefly describes the model.

---

## [Editor Report · Decision Letter 2]

18 Oct 2019

Dear Dr Michelot,

On behalf of my colleagues and the Academic Editor, Laura Machesky, I am pleased to inform you that we will be delighted to publish your Short Reports in PLOS Biology. 

Early Version

PRESS 

Kind regards,

Hannah Harwood

Publication Assistant, 

PLOS Biology

on behalf of

Di Jiang,

Associate Editor

PLOS Biology